# Zebrafish Model to Study Angiotensin II-Mediated Pathophysiology

**DOI:** 10.3390/biology10111177

**Published:** 2021-11-13

**Authors:** Bhagyashri Joshi, Ganesh Wagh, Harmandeep Kaur, Chinmoy Patra

**Affiliations:** 1Developmental Biology, Agharkar Research Institute, Pune 411004, India; bhagyashrijoshi@aripune.org (B.J.); gwagh@aripune.org (G.W.); 2Science and Technology, SP Pune University, Pune 411007, India; 3Keenan Research Centre for Biomedical Science and Li Ka Shing Knowledge Institute, St. Michael’s Hospital, Toronto, ON M5C 2T2, Canada; Harmandeep.kaur@unityhealth.to

**Keywords:** hypertension, AngII, zebrafish, cardiac remodeling, fibrosis

## Abstract

**Simple Summary:**

Hypertension or high blood pressure is a long-term incurable clinical condition characterized by persistent high blood pressure in arteries. Constant pressure overload in the heart leads to cardiac remodeling involving hypertrophy, alteration of gene expression, extracellular matrix molecule deposition, and cardiac fibrosis. In the long term, if left untreated, it can lead to myocardial infarction. Zebrafish is a freshwater fish that has been proven to be a very beneficial model system to study development, regeneration, and human diseases in recent years. We propose in this study zebrafish as a model to study Angiotensin II-mediated (AngII) multifactorial pathophysiology. AngII is an effector molecule of the Renin–Angiotensin system (RAS), which plays a crucial role in hypertension in mammals. In our study, AngII was injected at regular time intervals for a period of time. Our results show that, similar to mammals, AngII induces fibrotic gene expression, collagen deposition, cardiomyocyte hypertrophy, and cardiac cell proliferation. Thus, we propose that zebrafish can prove to be a valuable model to study AngII-RAS pathway-mediated pathophysiology.

**Abstract:**

Hypertension, a common chronic condition, may damage multiple organs, including the kidney, heart, and brain. Thus, it is essential to understand the pathology upon ectopic activation of the molecular pathways involved in mammalian hypertension to develop strategies to manage hypertension. Animal models play a crucial role in unraveling the disease pathophysiology by allowing incisive experimental procedures impossible in humans. Zebrafish, a small freshwater fish, have emerged as an important model system to study human diseases. The primary effector, Angiotensin II of the RAS pathway, regulates hemodynamic pressure overload mediated cardiovascular pathogenesis in mammals. There are various established mammalian models available to study pathophysiology in Angiotensin II-induced hypertension. Here, we have developed a zebrafish model to study pathogenesis by Angiotensin II. We find that intradermal Angiotensin II injection every 12 h can induce cardiac remodeling in seven days. We show that Angiotensin II injection in adult zebrafish causes cardiomyocyte hypertrophy and enhances cardiac cell proliferation. In addition, Angiotensin II induces ECM protein-coding gene expression and fibrosis in the cardiac ventricles. Thus, this study can conclude that Angiotensin II injection in zebrafish has similar implications as mammals, and zebrafish can be a model to study pathophysiology associated with AngII-RAS signaling.

## 1. Introduction

In hypertensive conditions, the force of blood against the blood vessel wall is much higher than the physiological condition both during the systolic and diastolic status of the heart [1,2,3]. If left unattended, it can affect multiple organs, resulting in myocardial infarction, stroke, and kidney damage [4,5,6]. A WHO report suggests that hypertension contributes to around 50% of the mortality associated with stroke and ischemic heart disease, accounting for about 7.5 million global deaths annually [7].

Although a handful of pharmacological treatments are available to manage hypertension in the present world [8,9], a remedy to cure hypertension is not available. Mechanical overload is divided into two categories: volume, and pressure overload [10]. Pressure overload causes hypertension and aortic stenosis, leading to concentric hypertrophy and cardiac remodeling [10]. Cardiac remodeling involves hypertrophy, alteration of gene expression, extracellular matrix (ECM) molecule deposition, and cardiac fibrosis, leading to myocardial infarction or heart attack [11]. In the hypertensive heart, the degree of fibrosis also varies from mild to severe. ECM structure is preserved in mild fibrosis despite increased collagen deposition and cardiomyocyte hypertrophy. Severe fibrosis accounts for extensive cardiomyocyte hypertrophy and cell death, with increased and disorganized ECM [12,13,14].

There are many established rodent models to study hypertension [15,16]. For example, spontaneously hypertensive rats (SHR) [17] and Dah1/salt-sensitive rats [18] are popular genetic strains to research hypertension. Another etiological factor used for inducing hypertension is the Renin–Angiotensin System (RAS), in which Angiotensin-II (AngII) plays a pivotal role [19,20]. Studies have shown RAS is an essential regulator of hypertension and a mediator of cardiac remodeling in both hypertensive patients [20,21,22,23] and hypertension-induced model organisms [24]. AngII is the effector protein of the RAS system. In the brain, AngII induces vasopressin secretion [25,26], which causes vasoconstriction [25,26]. In the adrenal gland, AngII releases aldosterone, which mediates reabsorption of sodium chloride (NaCl) and water in the kidneys [25,26], and in the heart, AngII, along with aldosterone, causes cardiovascular remodeling [27]. AngII binds to Angiotensin receptors, G protein-coupled receptors (GPCR), which are of two types: an AngII type1 receptor (AT1R) and an AngII type2 receptor (AT2R) [28,29,30]. AngII binds to AT1R in cardiac fibroblast and induces cardiac fibroblast proliferation and dedifferentiation to myofibroblast [31] through the deposition of ECM proteins, including collagens, fibronectin, and laminin, resulting in cardiac remodeling. While AT1R expression remains unchanged in fetal and neonatal hearts, AT2R is highly expressed in fetal hearts and decreases rapidly during postnatal life [32]. Activation of AT2R stimulates protein tyrosine phosphatase, which inactivates AT1R-activated mitogen-activated protein kinase [33,34]. Thus, AT2R counteracts AT1R functions.

Model organisms, including mice [35], rats [36], mini pigs [37], and rabbits [38], have been proven as an efficient way to study human pathophysiology [39]. Over the past decade, the zebrafish has emerged as a preferred model organism to study development, regeneration, human diseases, small molecule screening due to its rapid ex utero development, optical transparency, large clutch size, amenability for genetic manipulation, and a wide range of tissue regeneration [40]. Moreover, the zebrafish genome carries at least one orthologue of 70% protein-coding and 80% disease-causing human genes [40]. Thus, it is possible to study the involvement of conserved genes in human diseases using zebrafish disease models. Studies on the zebrafish have shown that it also has a functional RAS, similar to mammals [41]. In support of this, another study in zebrafish has shown that AngII infusion causes an increase in salt uptake in zebrafish larvae [42]. Recently Filice et al. have shown that long-term exposure to waterborne AngII leads to cardiac remodeling in adult zebrafish [43]. However, its function on cardiac cell proliferation, CM hypertrophy, and genetic regulation associated with CM hypertrophy and fibrosis were not explored or poorly studied. Thus, we propose developing a zebrafish model which may overcome the shortcomings of mammalian models to explore the pathophysiology upon intradermal delivery of AngII. We aim to establish the duration of AngII administration to develop AngII-mediated cardiac remodeling in adult zebrafish. Our study shows that intradermal injection of AngII for 7 days induces ECM coding and cardiac hypertrophy marker gene expression in cardiac ventricles. In agreement, AngII causes ectopic collagen deposition in the cardiac tissue, cardiomyocyte hypertrophy, and enhances cardiac cell proliferation. This study successfully shows that zebrafish have a similar response to AngII-induced cardiac pathophysiology that of mammals and establishes it as a model to study AngII-RAS signaling mediated pathophysiology.

## 2. Materials and Methods

### 2.1. Zebrafish Maintenance

4-to 8-month-old male transgenic *TgBAC*(*etv2:EGFP*)*ci1* [44] [hereafter *Tg*(*etv2:EGFP*)], *Tg*(*–5.1myl7:nDsRed2*)*f2* [45] [hereafter *Tg*(*myl7:nucDsRed*)], *Tg*(*myl7:EGFP-HsHRAS*)*^s883^* [46] [hereafter *Tg*(*myl7:HsHRAS-EGFP*)], and wild-type AB zebrafish were used in this study. The 5dpf embryos were transferred to a state-of-the-art zebrafish housing system (Techniplast, Italy) which maintains a temperature of 28 °C, pH ~7.5, and conductivity of ~500 µS/cm.

### 2.2. Ethics Statement

The guidelines recommended by the Committee for Control and Supervision of Experiments on Animals (CPCSEA), Government of India, were followed for zebrafish maintenance and experimentation. All the animal care procedures and protocols used in this study were approved by The Institute Animal Ethics Committee (IAEC).

### 2.3. Angiotensin II injections

Human AngII (sigma) was dissolved in PBS at 10 mg/mL PBS. AngII in PBS (test) or PBS (control) was administered intradermally in adult zebrafish using Hamilton’s syringe. One day before the first injection, the fishes were anesthetized in 0.02% tricaine, weighed, and similar weight fishes were kept in a group. An amount of 1.5 µg AngII/g of zebrafish was injected intradermally at an interval of 12 h for 7, 14, and 30 days (Figure 1A). For injection, fishes were anesthetized, immobilized dorsally into a wet foam holder (Figure 1B), and injected intradermally. Post-injection, fishes were transferred into fresh water and revived. Revived fishes were kept in a state-of-the-art zebrafish aquarium and fed thrice per day. In the morning and evening, fishes were fed at least half an hour after the injection. After the stipulated injection period, fishes were anesthetized and weighed before sacrificing and heart isolation.

### 2.4. qPCR and Gene Expression Analysis

Total RNA from cardiac ventricles from PBS or AngII injected animals was isolated using Trizol (Invitrogen) and chloroform: isopropanol (SRL). For every biological replicate, six ventricles were pooled, and 1–3 µg of RNA was reverse transcribed to cDNA using MMLV reverse transcriptase kit (Invitrogen) following the manufacturer’s instructions. qPCR was performed using cDNA with a PCR max real-time PCR detection system (Cole-Parmar, Staffordshire, UK). mRNA expression levels relative to *ef1α* (control) were analyzed by calculating the ∆Ct method. Gene expression levels were calculated as 2^−ΔCt^ mean values of output Ct obtained from the duplicates of qPCR assays for each of the three independent biological replicates of each condition. For each gene, the mean value of the expression level in control was considered 1, and data are plotted on a logarithmic scale. Primer sequences used in this study that obtained Ct values and calculated fold changes by the ΔCt method are mentioned in Appendix A, respectively.

### 2.5. Edu Incorporation Assay

Cell proliferation was assessed by EdU incorporation assay on the heart sections from AngII or PBS injected fishes. Fishes were anesthetized, and 10 μL of 10 mM EdU (Click-iT EdU, Invitrogen) was injected intraperitoneally 24 h before sacrificing the animal. Isolated hearts were fixed in 4% PFA in PBS for 20 min at room temperature, washed in PBS, and submerged in 30% sucrose until hearts settled in the bottom of the vial. Hearts were embedded in a tissue-freezing medium (OCT (optimal cutting temperature medium) (Leica, Germany)) and processed for cryosectioning (cryotome, Leica, Nusssloch, Germany). Then, 10 μm thin sections were taken on silane (HiMedia, India) coated slides (BOROSIL, India) and processed for EdU labeling according to the manufacturer’s instructions. DAPI (0.5 mg/mL water, Sigma) was used to detect DNA. For imaging, optical sections were captured using a Leica SP8 confocal microscope (Wetzlar, Germany). The z-stack images were processed using LAS X software (Leica, Wetzlar, Germany), and cell counting and area measurement were performed using ImageJ/Fiji software (NIH, Bethesda, MD, USA). A total of 5–9 hearts each from control and test animal were used, and at least four sections from each ventricle were analyzed for cell proliferation study.

### 2.6. Picrosirius Red Staining

Hearts were subjected to serial dehydration using ethanol, water grades, and xylene. Tissues were embedded and aligned in paraffin wax for sectioning. Thin sections of 10 μm were taken with a microtome (Leica, Nussloch, Germany), mounted on silane-coated glass slides, and dried at 37 °C overnight. Tissue sections were rehydrated, and picrosirius red staining was performed following a standard protocol [47]. After staining, sections were dehydrated to remove excess stain and mounted with a xylene-based mountant Entellan (Sigma), and bright-field images were captured using a Leica microscope (DM5500B, Wetzlar, Germany). The area covered by picrosirius red-stained collagen was measured following the ImageJ tutorial (https://imagej.nih.gov/ij/docs/examples/stained-sections/index.html, accessed on 3 September 2021) “Quantifying stained liver tissue” [48]. In brief, first, the images were converted to grayscale, followed by isolating the red-stained collagen using thresholding. The threshold tissue area was selected and measured. For quantification, 4–6 hearts each from control and test animals were used, and at least 8 sections from each ventricle were analyzed.

### 2.7. Cardiomyocyte Cross-Section Area Analysis

*Tg*(*myl7:HsHRAS-EGFP*) transgenic zebrafish, which express EGFP in the CMs plasma membrane, was used for measuring the CM cross-sectional area. Immunohistochemical analysis was performed on the 10 μm thin transverse sections (cryotome, Leica) of ventricles as described previously [49]. In brief, PBS-washed tissue sections were re-fixed, permeabilized, and blocked for 1 h in a blocking solution [5% goat serum (MP Biomedicals) and 0.2% Tween-20 in PBS]. After blocking, primary antibody treatment was conducted overnight at 4 °C in blocking solutions. Rabbit anti-GFP, 1:400 (Novus Biologicals, NB600-308) primary antibody was used in this study. Primary immune complexes were detected by AlexaFluor^488^-conjugated secondary antibody (1:400; Molecular Probes, A11034). DAPI (0.5 mg/mL, Sigma) was used to detect DNA. Confocal sections were captured using a Leica SP8 confocal microscope (Leica, Wetzlar, Germany). The cross-sectional area of the CMs in the cortical zone was measured from single plane optical sections using ImageJ (NIH, USA). A total of 5 hearts each from PBS or AngII injected animals were used for analysis. An area of 50 CMs were measured from each section, and for every heart, 2 sections were used. The data were quantified using a non-parametric *t*-test.

### 2.8. Statistical Analysis

Statistical differences of qPCR relative expression data were analyzed using a two-tailed Student’s *t*-test. An unpaired *t*-test with Welch’s correction was employed for the statistical analysis for weight loss, the fibrotic area in the cardiac ventricles, cell proliferation, and hypertrophy. The changes in the analysis were considered to be statistically significant if *p* < 0.05. Values were represented as mean ± s.d. or mean ± s.e.m., and all the statistical analysis was performed using GraphPad Prism7 software.

## 3. Results

### 3.1. AngII Induces Cardiac Hypertrophy and Overall Weight Loss in Adult Zebrafish

Previous studies have shown that AngII employment through an osmotic pump in rodents causes cardiac hypertrophy [50,51] and remodeling [52]. Since the osmotic pump delivery system is unavailable for zebrafish, we opted for intradermal injection, which ensures relatively slow drug release in the body and bloodstream from the injection site [53]. In rodents, an osmotic pump-based delivery of ~1 µg AngII/g body weight in 12 h for 2–3 weeks leads to pressure overload-induced cardiac hypertrophy [54,55]. Since we planned to inject every 12 h, in the place of osmotic pump-based continuous delivery, we employed a 50% higher dose than rats. Thus, to explore the effect of AngII on adult zebrafish, 1.5 µg AngII in PBS/g of zebrafish was injected intradermally every 12 h for a stipulated time (Figure 1A,B). Studies in rodents have reported AngII treatment for a maximum of up to 21 days, and none of those studies reported mortality [54,55,56,57]. Similarly, we found 100% viability in AngII or PBS injected zebrafish until 21 days post-injection (dpi) (*n* = 18 each from three independent experiments). However, around ~23% and ~67% mortality was observed in AngII injected zebrafish at 23 and 30 dpi, respectively (*n* = 18 each from three independent experiments) (Figure 1C).

Since we observed mortality post 21 dpi, we sought to analyze the effect of AngII in the hearts at 7 and 14 dpi. Size and age-matched males were used for control or AngII injection in each batch. While at 7 dpi, AngII injected animals looked similar to control (Figure 1D), the overall size of cardiac ventricles was larger in AngII injected animals compared to the control group (Figure 1E). The ratio of the 2D surface area of ventricle and body length was quantitated for each animal (*n* = 5 from two independent experiments). The analysis showed that cardiac ventricles of the AngII injected animals were around 16% larger than the PBS injected control (Figure 1F), suggesting AngII induces cardiac hypertrophy in zebrafish.

In rodents, an increased level of AngII causes weight loss [58,59]. Thus, we analyzed weight loss in AngII or PBS injected zebrafish at 7 and 14 dpi. Fishes were weighed and grouped according to their weight before the initiation of the experiment. The weight difference between pre-injection (day 0) and after 7 or 14 dpi was calculated for each animal. Weight difference (bodyweight on day 0 to bodyweight at 7 or 14 dpi) in mg/100 mg fish body weight showed that PBS injected fishes gained ~2–5% of the day 0 weight at 7 and 14 dpi. In contrast, AngII injected fishes had undergone ~8.5%, and ~11% loss of the day 0 weight at 7 dpi and 14 dpi, respectively (*n* = 14 each from three independent experiments) (Figure 1G), suggesting AngII induces weight loss in zebrafish. AngII application in rodents leads to weight loss via suppressing circulating IGF-I expression [58]. Our qPCR analysis showed decreased *igf1* expression in the cardiac tissue of AngII injected zebrafish (Figure 1H). Thus, our experiments suggest an intradermal injection of 1.5 µg AngII/g in zebrafish every 12 h is effective in causing cardiac hypertrophy in 7 days and showed weight loss in zebrafish similar to rodents.

### 3.2. AngII Injection Leads to Fibrotic Gene Expression and Scarring in Zebrafish Hearts

Hypertension leads to collagenous fibrosis in human hearts [3]. Cardiac fibrosis was also observed in AngII-induced hypertension in mice [60,61,62]. Since our model exhibited AngII-induced cardiac hypertrophy at 7 dpi, we further explored collagenous scars in cardiac ventricles at 7 and 14 dpi. Picrosirius red-stained sagittal wax sections of the ventricles from AngII injected 7 dpi, and 14 dpi animals showed interstitial fibrosis in the cardiac tissue compared to control hearts (Figure 2A). For analysis, bright-field images were quantified for picrosirius red-stained collagen enriched area/mm^2^ cardiac tissue. Quantitative analysis showed that compared to the PBS injected control, in the heart sections from AngII injected animals, ~85% and ~83% more area was collagen enriched at 7 and 14 dpi, respectively (*n* = 5 each from two independent experiments) (Figure 2B,C). Thus, our data indicate that AngII induces cardiac fibrosis in zebrafish.

It is well established that collagen type I and III are upregulated in AngII-induced fibrosis in mammals [61,62,63]. Besides the collagens [62,63]; *FN* [61], *TNC* [64], osteopontin (*SPP1*) [65,66], and *LOX*, a molecule involved in collagen deposition and cross-linking [67] are molecular markers of cardiac fibrosis in mammals. As we observed induction of cardiac fibrosis upon AngII injection, we sought to explore collagen and non-collagen coding fibrotic gene expression levels in the cardiac ventricles. qPCR analysis showed that the expression levels of the ECM protein gene *col1a1a*, *col1a2*, *col1a1b*, *fn1a*, *fn1b*, *tnc*, and *loxa* were increased by ~1.7 to 7 folds; those of *col13a1*, *col15a1b*, and *ccn2a* exhibited a trend in increased expression in cardiac ventricles of AngII injected animals at 7 dpi (Figure 2D). Downregulated *col12a1b* expression by ~5 folds was observed in the hearts of AngII injected fish relative to control, whereas the expression of *col8a1a*, *col2a1b*, and *col12a1b* was not regulated at 7 dpi (Figure 2D). Thus, our data suggest that AngII induces cardiac fibrotic gene expression and cardiac fibrosis in zebrafish alike in mammals.

### 3.3. AngII Injection Induces Cardiomyocyte Hypertrophy in Zebrafish

In mammals, AngII induces CM hypertrophy, fibrosis, and thus cardiac remodeling [31,50,52]. We observed cardiac hypertrophy in the AngII injected zebrafish (Figure 1E). Therefore, we explored the effect of AngII on the size of CMs at 7 dpi. The cross-section area of CMs was quantified using *Tg*(*myl7:HsHRAS-EGFP*) [46] animals that express EGFP in the CMs plasma membrane, facilitating analyzing CM morphology. Transverse sections through the cardiac ventricle were stained with anti-GFP and scanned under a confocal microscope. Single plane optical images showed a bigger cross-sectional area of the CMs in the cortical myocardium of the heart from AngII injected zebrafish compared to PBS injected control (Figure 3A). Quantitative analysis showed that CMs from AngII injected animals are ~2 times larger than the control hearts at 7 dpi (*n* = 5 from two independent experiments) (Figure 3B), suggesting AngII induces CM hypertrophy in zebrafish.

In mammals, *NPPA* and *MYH7* are induced in hypertrophic CMs [68,69]. In zebrafish, there are two paralogs of *MYH7*: *myh7ba* and *myh7bb*. qPCR analysis showed that *nppa*, *myh6*, and *myh7ba * were increased by ~2 fold, while *myh7bb* showed a trend in increased expression in the cardiac ventricles from AngII injected zebrafish compared to control hearts (Figure 3C). Overall, our CM morphology and marker gene expression analysis showed that AngII induces CM hypertrophy in zebrafish.

### 3.4. AngII Injection Promotes Cardiac Cell Proliferation in Adult Zebrafish

In vitro studies support that AngII positively regulates fibroblasts [34] and endothelial cell [70] proliferation. *Similarly*, AngII induces cardiac fibroblast proliferation [34,60,61,62] and microvascular growth [71] in mammals. We therefore performed a 5-ethynyl-2′-deoxyuridine (EdU) incorporation assay to quantify cardiac cell proliferation indices in AngII or PBS injected zebrafish (Figure 4A). To determine the proliferation indices of different cardiac cell types, we utilized transgenic zebrafish; *Tg*(*myl7:nucDsRed*), which express RFP in cardiomyocyte nuclei and *Tg*(*etv2:EGFP*), in which EGFP is localized in the endothelial cells. Very few EdU positive cardiac cells were found in control hearts (Figure 4B,C). EdU incorporation by CMs, ECs, non-CMs and non-ECs nuclei was quantified in sagittal heart sections at 7 dpi. Our assessments revealed ~40 fold enhancement of overall cardiac cell proliferation in AngII injected animals compared to control hearts (Figure 4B–D) (*n* = 7 each from three independent experiments). We found in control hearts ~0.5, ~0.1, and ~28 EdU^+^ CMs, ECs, and non-CMs, respectively, in per mm^2^ cardiac tissue at 7 dpi (Figure 4). In comparison, ~8 EdU+ CMs, ~100 EdU^+^ ECs, and ~1070 EdU^+^ non-CMs were detected per mm^2^ of cardiac tissue in AngII injected animals at 7 dpi (Figure 4), suggesting AngII enhances cardiac cell proliferation. This result indicates that AngII enhances EC, CM, and non-CM/EC proliferation in adult zebrafish hearts.

## 4. Discussion

Our study aimed to develop a zebrafish model to study RAS-AngII signaling, a crucial player in hypertension mediated pathophysiology in mammals. We have shown that intradermal AngII injections every 12 h for 7 days in zebrafish causes cardiac hypertrophy, cardiac fibrosis, and induction of cardiac cell proliferation, which are characteristic responses to pressure overload in mammalian hypertension.

In this study, AngII injection does not show any mortality until 21 dpi. However, AngII administration beyond 21 dpi showed high mortality. In rodents, AngII is administered using a subcutaneously implanted osmotic pump to ensure slow release over a stipulated time [56,72]. As the continuous release of drug can be provided by an osmotic pump, a relatively low and constant level of AngII can be maintained. As per standard protocol in rodents, an osmotic pump releases 0.5–1.5 µg/kg/min for 2–3 weeks, which does not lead to induction of mortality until 21 days [54,55,56,57,73]. To ensure the effectiveness of AngII, in this study, we used bolus injections of AngII, which likely resulted in a rapid increase in blood concentration of AngII, and maintained a high concentration for a substantial time after each injection. It could be the reason for the adverse effects of AngII, leading to increased mortality in long-term treatment. Developing a sustained release dosage form, such as hydrogel-based AngII delivery or micro-osmotic pump to keep a steady plasma concentration of AngII over time may overcome the mortality issue in long-term experimental procedures in adult zebrafish.

Brink et al. and Shen C et al. have shown that AngII administration leads to weight loss in rodents [58,73]. Filice et al. have shown that waterborne AngII does not induce weight loss even after 8 weeks of treatment [43]. In contrast, such as with rodents, we found a significant reduction in weight in AngII injected fishes. Published evidence indicates the weight loss in rodents upon AngII application is due to muscle wasting via anorexia [59,73] and suppression of circulating IGF-I and IGF-I pathway [58], which is necessary for muscle building. We identified decreased *igf1* transcripts in the cardiac ventricles of AngII injected zebrafish. Thus, the AngII-induced weight loss mechanism could be conserved between mammals and zebrafish, but it needs further exploration.

Studies have shown upregulation of type I, type III, and type IV collagens in AngII-induced fibrosis in rodents [61,62,63]. In line with the mouse data, we found fibrillar collagen coding genes, including *col1a1a*, *col1a2*, and *col1a1b*, and fibril-associated collagen coding gene *col15a1b* were upregulated in the ventricle of AngII employed zebrafish. In contrast, *col12a1b*, fibril-associated collagen was found to be decreased in the AngII injected animals. Although there is no report about the *Col12a1* expression in pressure overload mammalian models, it is known to be induced in injury response cardiac fibrosis in mouse and zebrafish [74] hearts. We believe this suggests that *col12a1b* expression is context-specific in cardiac fibrotic tissue. In mice, increased expression of *Loxa* [67], *Tnc* [64], and *Spp1* [65,66] is associated with the processes involved in AngII-induced cardiac fibrosis. Similar to mammals, we found increased expressions of *fn1a*, *fn1b*, *loxa,* and *tnc* transcripts in AngII-induced zebrafish hearts, while *spp1* expression remained unaltered. In contrast, our previous study identified ~15 fold increased expression of *spp1* in injured zebrafish hearts [75]. These findings put forward that regulation of *spp1* expression in cardiac fibrosis is context dependent. Overall, gene expression analyses indicate that similar genetic regulation exists between mammals and zebrafish in AngII-induced fibrosis.

In mammals, pressure overload on the atrial and ventricular walls induced the expression of embryonic genes *NPPA* [68] and *MYH7* [76], which are not expressed in healthy adult hearts [68,69,77]. In agreement with the mammalian data, we found that *nppa*, *myh7ba*, and *myh7bb* either significantly induced or showed a trend of increased expression in the AngII injected zebrafish cardiac tissue, suggesting zebrafish maintained the genetic signature of AngII-induced cardiac hypertrophy similar to mammals.

In line with the mammalian data, our EdU incorporation assay suggests that AngII induces cardiac cell proliferation and a majority of the EdU positive cells are non-CMs and non-ECs (Figure 4). Presumably, the majority of the non-CMs and non-ECs are cardiac fibroblasts [49,78,79]. We found most of the EdU^+^/DAPI^+^ non-CMs and non-ECs are in clusters in the hearts of AngII injected animals (Figure 4B,C). Thus, most likely, in zebrafish hearts, AngII induces cardiac fibroblast proliferation similar to in mammals, but it needs to be further elucidated. *An* in vitro study showed that AngII does not induce mammalian CM proliferation [31]. In contrast, AngII promotes CM proliferation in zebrafish (Figure 4). It might be because adult zebrafish CMs have a substantial proliferative capacity [80]. Sartore et al. have shown increased *Myh6* expression in injured rat skeletal muscle cells [69]. In regenerating zebrafish hearts, embryonic *myh7* positive undifferentiated CMs have been identified in the border region of the injured zebrafish heart [81]. These data indicate embryonic gene activation during muscle regeneration. We found increased expression of embryonic genes, including *myh6*, *myh7ba,* and *myh7bb* in AngII injected animals. Thus, it is possible that in zebrafish, AngII mediated pressure load induces CM dedifferentiation, which promotes CM cell cycle reentry, but this needs to be further explored.

## 5. Conclusions

Our study shows that induction of AngII in zebrafish leads to cardiac remodeling. As in rodents, AngII infusion induces fibrosis, cardiac hypertrophy, fibroblast proliferation, and endothelial cell proliferation in zebrafish. However, unlike mammals, in zebrafish we observed that AngII promotes cardiomyocyte proliferation. Overall, we showed that cellular and molecular processes involved in AngII-induced cardiac remodeling are conserved between mammals and zebrafish. Thus, we propose that zebrafish could be a sound model system to study RAS-AngII signaling mediated pathophysiology and may facilitate the research to identify novel strategies to manage multifactorial pathophysiology associated with the RAS-AngII pathway.

## Figures and Tables

**Figure 1 biology-10-01177-f001:**
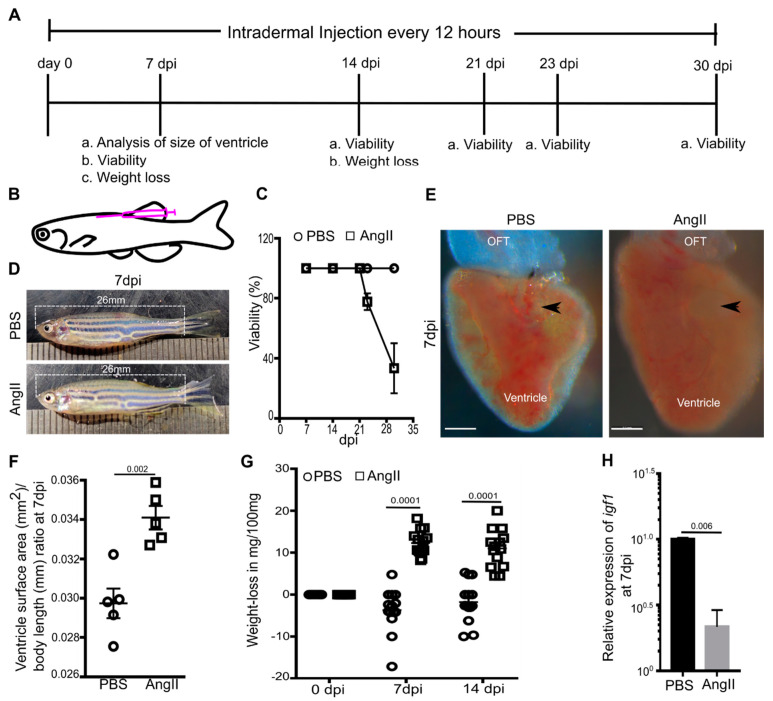
Intradermal Angiotensin II injection induces cardiac hypertrophy in adult zebrafish. (**A**,**B**) Diagrams showing experimental procedure (**A**) and site of intradermal injection (**B**). (**C**) Survival assay assessed the viability of the PBS or AngII injected animals at different time points (*n* = 18 each from 3 independent experiments). Error bars indicate the mean ± s.d. (**D**) Images of PBS or AngII injected zebrafish at 7 dpi. (**E**) Bright-field images of cardiac ventricles isolated from PBS or AngII injected animals at 7 dpi. Arrowheads indicate the atrioventricular canal. Scale bars, 200 µm. (**F**) Statistical analysis of the ratio of the 2D surface area of the ventricle and body length of individual fish at 7 dpi (*n* = 5 each from 2 independent experiments). (**G**) Quantitative analysis of weight loss in 14 PBS injected and 14 AngII injected zebrafish from 3 independent experiments. (**H**) Quantitative analysis of the expression of *igf1* in cardiac ventricles from PBS or AngII injected animals at 7 dpi (*n* = 3, each sample represents a pool of 6 hearts). Error bars indicate the mean ± s.e.m. dpi: days post-injection; OFT, outflow tract. Significant, *p* < 0.05; non-significant, *p* ≥ 0.05.

**Figure 2 biology-10-01177-f002:**
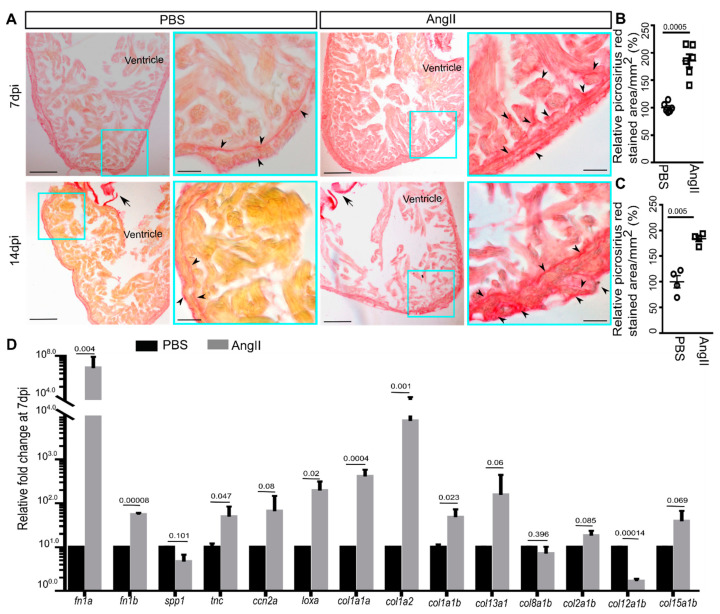
Angiotensin II injection leads to cardiac fibrosis. (**A**) Bright-field images of picrosirius red-stained sagittal wax sections of cardiac ventricles from PBS or AngII injected animals at 7 dpi and 14 dpi (collagen, red). Arrowheads indicate collagenous scarring in the ventricular tissue and arrows indicate cardiac valves. Scale, low zoom: 100 µm, high zoom: 20 µm. (**B**,**C**) Quantitative analysis of the picrosirius red-stained area in the heart sections of PBS or AngII injected animals at 7 dpi (6 each from 2 independent experiments) (**B**) and 14 dpi (4 each from 2 independent experiments) (**C**). From each heart, 8–10 sections were analyzed for quantification. The mean of control was considered to be 100%. (**D**) Quantitative analysis of collagen and fibrosis marker gene expression in cardiac ventricles from PBS or AngII injected animals at 7 dpi (*n* = 3, each sample represents a pool of 6 hearts). Error bars indicate the mean ± s.e.m. values in (**D**) are normalized to the mean of the PBS control. Dpi: days post-injection. Significant, *p* < 0.05; non-significant, *p* ≥ 0.05.

**Figure 3 biology-10-01177-f003:**
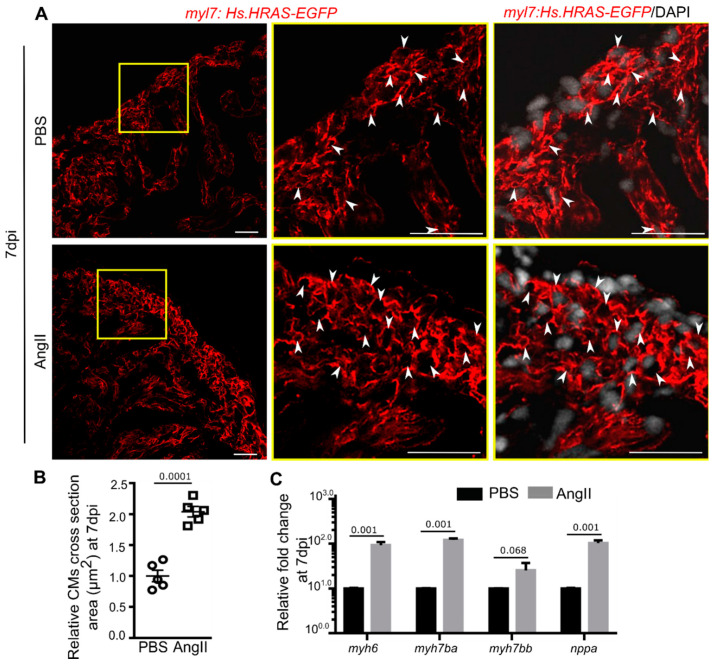
Angiotensin II induces cardiomyocyte hypertrophy in adult zebrafish. (**A**) Confocal optical sections of transverse cryosections of hearts isolated from PBS or AngII injected animals expressing EGFP in CM plasma membrane (red) and stained with DAPI (white; marks all nuclei). White arrowheads point to the plasma membrane of cross-sectioned CMs. (**B**) The dot plot represents the relative cross-sectional area of CMs at 7 dpi (*n* = 5 each from two independent experiments). At least 50 CMs from each section and 2 sections from each heart were included in the analysis. The mean of the cross-sectional area of CMs from PBS injected hearts was set to 1. (**C**) Quantitative analysis of the expression of hypertrophy marker genes in cardiac ventricles from PBS or AngII injected animals at 7 dpi (*n* = 3, each sample represents a pool of 6 hearts). Error bars indicate the mean ± s.e.m. values in C were normalized to the mean of PBS control. Dpi: days post-injection. Significant, *p* < 0.05; non-significant, *p* ≥ 0.05. Scale: 20 µm.

**Figure 4 biology-10-01177-f004:**
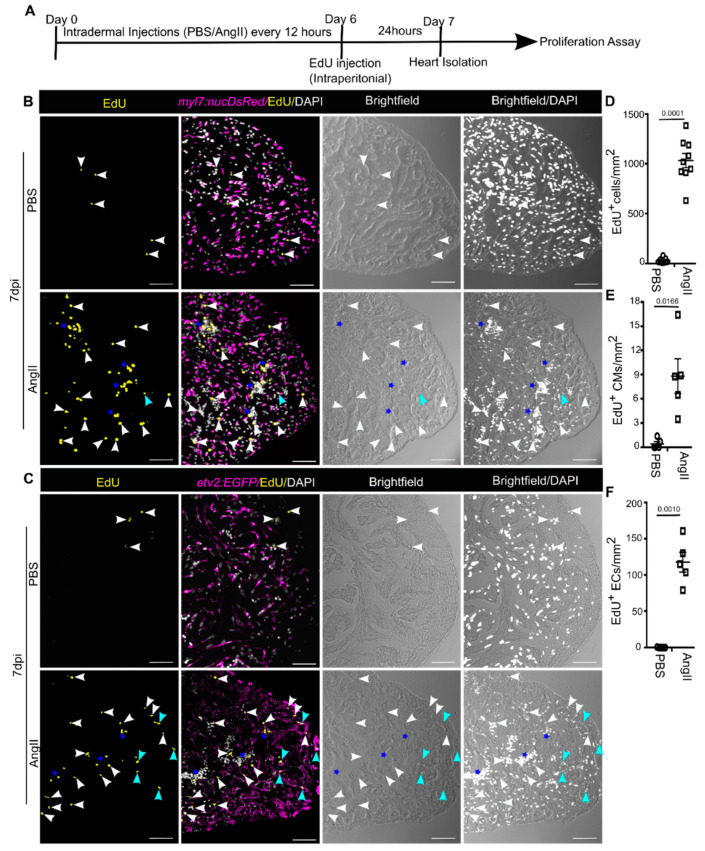
Angiotensin II injection enhances cardiac cell proliferation. (**A**) Diagram showing the experimental procedure. (**B**) Representative single plane confocal images of sagittal cryosections of hearts isolated from PBS or AngII injected animals expressing DsRed in CM nuclei (magenta), stained for EdU (yellow: marks proliferating cells), and stained with DAPI (white: marks all nuclei). White and blue arrowheads point to EdU^+/^DAPI^+^ cells and EdU^+/^DAPI^+^/DsRed^+^ cells, respectively. Stars indicate cell clusters. (**C**) Confocal optical sections of sagittal cryosections of hearts isolated from PBS or AngII injected animals expressing EGFP in ECs (magenta), stained for EdU (yellow: marks proliferating cells), and stained with DAPI (white: marks all nuclei). White and blue arrowheads point to EdU^+/^DAPI^+^ cells and EdU^+/^DAPI^+^/EGFP^+^ cells, respectively. Stars indicate cell clusters. (**D**,**E**) Cardiac cell (*n* = 9 each) (**D**) and CM (*n* = 5 each) (**E**) proliferation was quantified in cardiac ventricles from PBS or AngII injected animals at 7 dpi. (**F**) EC proliferation was quantified in cardiac ventricles from PBS or AngII injected animals at 7 dpi (*n* = 5 each). At least 2 sagittal sections of each heart were analyzed for quantification in (**D**–**F**). Error bars indicate the mean ± s.e.m. Dpi: days post-injection. Significant, *p* < 0.05; non-significant, *p* ≥ 0.05. Scale: 50 µm.

## Data Availability

All data are available from the corresponding author upon reasonable request.

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
