# Peer review of "Zebrafish Model to Study Angiotensin II-Mediated Pathophysiology"

_biology, 2021, doi:10.3390/biology10111177_

Round 1

Reviewer 1 Report

In this manuscript, the authors aim at establishing the hypertension disease model in zebrafish adults. They administer Angiotensin II via intradermal injection, and follow the effects over the course of several days. They follow the phenotypic changes that have been previously described in rodent models, including weight loss, increase in cardiac hyperthrophic markers and proliferation of non-cardiomyocytes. With their approach, the authors manage to establish that administration of Angiotensin II for up to day 14 post injection leads to hypertension-like phenotype in adult zebrafish and can serve as a timely approach to study this multifactorial disease condition in zebrafish.

Major comments:

On what bases the authors select the dose? This should be clarified.

The authors only test the duration of AngII administration. They do not test different modes of administration or different concentrations of AngII. Line 85: it should be edited accordingly throughout.

In qPCR analysis, are directly the delta-Ct values statistically tested? If not then the two tail t-test is not correct as the relative change is compared to 1. How are error bars generated if the control group is set to 1? As a suggestion, delta-Ct values are be tested directly, and the data are plotted on logarithmic scale with the control set to 0.

Minor

For wight measurements, use g instead of gm

1F, y axis, typo in area

Fig4c: transgenic line is labeled as fli1:EGFP but in methods and text it is referred to as etv2:EGFP, this should be unified

Italics or bolt should not be used in discussion for emphasis

Line 263: …”col12a1b …”  is this a typo?

Author Response

We thank you for your time to thoroughly review our manuscript and your insightful comments. Please find your comments addressed point by point (in blue font) in our reply below. Alterations to the original manuscript are marked up using the “Track Changes” function in the revised version of our manuscript.

*Major comments:

On what basis the authors select the dose? This should be clarified.

We thank you for pointing this out. We have included the information and made the below changes to the manuscript.

Changes to the manuscript:

  1. Addition to Results: ‘In rodents, an osmotic pump-based delivery of ~1µg AngII/g body weight in 12 hours for 2-3 weeks, leading to pressure overload-induced cardiac hypertrophy[53,54]. Since we planned to inject in every 12 hours, in the place of osmotic pump-based continuous delivery, we have employed a 50% higher dose than rats. Thus, to explore the effect of AngII on adult zebrafish, 1.5µg AngII in PBS/g of zebrafish was injected intradermally every 12 hours for a stipulated time (Fig. 1A, B). (Line# 216-221).

The authors only test the duration of AngII administration. They do not test different modes of administration or different concentrations of AngII. Line 85: it should be edited accordingly throughout.

We made suggested corrections (Line#108-109).

In qPCR analysis, are directly the delta-Ct values statistically tested? If not then the two tail t-test is not correct as the relative change is compared to 1. How are error bars generated if the control group is set to 1? As a suggestion, delta-Ct values are be tested directly, and the data are plotted on logarithmic scale with the control set to 0.

We thank you for pointing this out and apologize for not clearly describing the analysis. Yes, in the qPCR study, we have directly statistically tested the delta-Ct values. The mean of delta-Ct values of the control samples (n=3) was considered 1. Fold changes in the test samples were calculated by the delta-Ct method. Where ever three values of the controls are close to each other, error bars are not visible. 

Changes to the manuscript:

  1. Addition to Methods: “Gene expression levels were calculated as 2-ΔCt mean values of output Ct obtained from the duplicates of qPCR assays for each of three independent biological replicates of each condition. For each gene, the mean value of the expression level in control was considered 1, and data are plotted on a logarithmic scale. Primer sequences used in this study, obtained Ct values, and calculated fold changes by the ΔCt method are mentioned in Tables S1, S2, and S3, respectively.” (Line# 151-156).
  2. Addition to Figures: (i) We have plotted the data on a logarithmic scale for all qPCR-related graphs.

(ii) We have now provided gene expression fold change values in Table-S3.

*Minor Comments

For wight measurements, use g instead of gm

We made suggested corrections throughout the manuscript.

1F, y axis, typo in area

We made suggested corrections.

Fig4c: transgenic line is labeled as fli1:EGFP but in methods and text it is referred to as etv2:EGFP, this should be unified.

We thank the reviewer for pointing this out and apologize for this mistake. In this study, we have used the etv2:EGFP line only. We made the corrections.

Italics or bolt should not be used in discussion for emphasis

We made suggested corrections.

Line 263: …”col12a1b …”  is this a typo?

It is not a typo.

Reviewer 2 Report

This article provides a very nice introduction and interesting view on an untypical model of cardiac remodeling caused by pressure overload in an adult zebrafish. Although this article is written clearly I have some comments.

First of all, I recommend carefully check English. In the text are some typos and errors in the word order in the sentences. Then in some cases, can sentence have a different meaning.

Some shortcuts explanations are missing in the text. Also in the shortcuts are typos: IFG to IGF?

Some information about ATR2 should be added.

I recommend authors to focus on the part- material, and methods. There are some repeated sentences and also some sentences that are unsuitable.

In the method part is not united information about animal numbers in groups ( only in the results). This information should be added.

Not everywhere is information about glasses slides per group for microscopy.

Why authors mention in the legends of results the type of statistical evaluation. It is written in the methods part.

The authors indicate 3 experimental groups in their study, but mostly results only from 1 group are presented. These results should be added or should be reorganized the presentation of the results, where it will be clearly distinguished from which group (mostly from 7 days) are results. 

Explanation of arrowhead is missing in Figure 2.

In the text ist replaced IF to 1H.

The discussion part is more descriptive.

Author Response

We thank you for your time to thoroughly review our manuscript and your insightful comments. Please find your comments addressed point by point (in blue font) in our reply below. Alterations to the original manuscript are marked up using the “Track Changes” function in the revised version of our manuscript.

First of all, I recommend carefully check English. In the text are some typos and errors in the word order in the sentences. Then in some cases, can sentence have a different meaning. Some shortcuts explanations are missing in the text. Also in the shortcuts are typos: IFG to IGF?

We thank you for pointing these out. We went through the manuscript and made necessary corrections wherever needed.

Some information about ATR2 should be added.

We thank you for the suggestion. As suggested, we have provided the information.

Changes to the manuscript:

  1. Addition to Introduction: “While AT1R expression remains unchanged in fetal and neonatal hearts, AT2R is highly expressed in fetal hearts and decreases rapidly during postnatal life[32]. Activation of AT2R stimulates protein tyrosine phosphatase, which inactivates AT1R–activated mitogen-activated protein kinase[33,34]. Thus, AT2R counteracts AT1R functions.” (Line# 86-90).

I recommend authors to focus on the part- material, and methods. There are some repeated sentences and also some sentences that are unsuitable.

We went through the Materials and Methods section and made necessary corrections as suggested.

In the method part is not united information about animal numbers in groups ( only in the results). This information should be added.

We have included the information in the Materials and Methods section.

Not everywhere is information about glasses slides per group for microscopy.

In the Present manuscript, we have included the animal numbers and number of tissue sections from each heart were analyzed in the Materials and Methods section.

Why authors mention in the legends of results the type of statistical evaluation. It is written in the methods part.

As suggested, we have removed this information from legends.

The authors indicate 3 experimental groups in their study, but mostly results only from 1 group are presented. These results should be added or should be reorganized the presentation of the results, where it will be clearly distinguished from which group (mostly from 7 days) are results.

As suggested, we have indicated the time points in every figure, figure legends, and results. 

Explanation of arrowhead is missing in Figure 2.

We thank you for pointing this out. We have included the information in the Figure legend. (Line#279-280)

In the text ist replaced 1F to 1H.

We made the suggested correction. (Line#261)

The discussion part is more descriptive.

We went through the Discussion and removed some descriptive parts from the Discussion.
